# Three-Dimensional Division of Visible Light Communication Irradiation Area

**DOI:** 10.3390/s23010094

**Published:** 2022-12-22

**Authors:** Yang Zhou, Yuanzhi Deng, Huajie Wen, Liting Chen, Gang Xu

**Affiliations:** 1The College of Mechatronics and Control Engineering, Shenzhen University, Shenzhen 518060, China; 2The Shenzhen Key Laboratory of Urban Rail Transit, Shenzhen Technology University, Shenzhen 518118, China

**Keywords:** bit error rate, intensity modulation and direct detection technique, line-of-sight channels, visible light communication, volume ratio

## Abstract

In this article, we divide the irradiated area of visible light communication (VLC) into three parts, according to the influence of diffuse reflection, the irradiance half angle at the source and the communication distance on VLC. We present a volume ratio method to quantitatively analyze each divided part. In this work, based on the Lambertian reflection model of the VLC system in line-of-sight channels, five factors affecting the VLC performance are compared and discussed. A VLC system of a single white-light-emitting diode in a 10 m line-of-sight channel indoors is designed by using the intensity modulation and direct detection technique, and a three-dimensional model of the irradiated area is established.By comparing the distribution of the bit error rate (BER) of the optical signal at different lampshade heights, the volume ratio method is used to calculate the volume percentage of the three irradiated areas. The experimental results show that area II with a volume ratio greater than 50% is the best signal receiving area when compared with areas I and III, having a volume ratio in the range 20∼30%.

## 1. Introduction

Visible light communication (VLC) [1], as a new optical wireless communication method, has been explored for a wide range of applications, for example, indoor illumination and communication [2,3,4,5], indoor positioning [6,7,8,9,10], vehicle-to-vehicle communication [11,12] and vehicle–road coordination [13,14]. It is the most potential supplement to 5G wireless communication [15,16,17].

Compared with traditional wireless communication methods, the study of VLC systems in line-of-sight (LOS) channels mainly focuses on the improvement of communication rate and the optimization of modulation methods [18]. A feasible solution to improve the communication speed of VLC is presented in [19,20,21,22,23]. Wei et al. [23] proposed a VLC solution with a communication rate of up to 40.665 Gbps. The effects of various signal modulation methods, such as on–off keying [24,25], orthogonal frequency division multiplexing [26,27,28], quadrature amplitude modulation [29], generalized spatial modulation [30], and intensity modulation and direct detection (IM/DD) [31,32], on the performance of VLC systems have been investigated. Both single signal modulation methods and the hybrid modulation methods mentioned above improve the communication performance of VLC. The communication rate of the VLC system and the quality of the signal modulation methods are important indicators in evaluating the performance of a communication system.

In practice, in addition to the above indicators, irradiation area, optical power, and signal propagation angle are also considered—especially when the positions of the transmitter and the receiver are not fixed. The change in the position of the transceiver directly influences the above indicators, which in turn affects the quality of the communication performance. Therefore, when the parameters of the transmitter remain unchanged, ensuring that there is a sufficiently large receivable area for optical signals in the communication channel has a great impact on improving the communication performance of VLC systems.

The communication channel is an important part of the whole VLC system. Related studies have shown that the improvement of communication channels helps to enhance the communication performance of the system. Both LOS channels and non-LOS (NLOS) channels are analyzed in the literature [33,34,35]. Mohanna et al. [33] analyzed the effect of reflected light from walls on the receiver side by simulating a LOS channel and an NLOS channel. By arranging multiple LEDs in both LOS and NLOS channels, Priyanka et al. [34] improved the uniformity of light illumination within the communication channel and mitigated the effect of inter-symbol interference. Long [35] researched the performance evaluation of a VLC system with various FOVs, elevation and azimuth receivers by simulating LOS and NLOS models. A mixed-integer linear programming model was developed.

At the same time, some researchers have turned their attention to VLC channels in different application scenarios. Caputo et al. [36] introduced a VLC channel for I2V communication. The corresponding propagation model was established by linear regression. The proposed model showed higher accuracy than the traditional Lambertian model. Ding et al. [37] investigated the VLC channel’s characteristic evolution from the conventional Lambertian beam to typical non-Lambertian beams. Studies have shown that side-emitter beams provide a more uniform capacity distribution in a distributed transmitter configuration. Zhu et al. [38] proposed a three-dimensional space-time-frequency non-stationary-geometry-based stochastic model for indoor VLC channels. The accuracy and practicality of the model were verified by simulation experiments. Palacios et al. [39] proposed a hemispherical three-dimensional dust particle distribution model based on an underground mining scenario. By comparing with other models, the proposed model was found to be more accurate and realistic in terms of metrics assessment. Li et al. [40] constructed a neural-network-based transceiver for compensating the effect of variable inter-symbol interference in a VLC system. The results showed that the neural-network-based transceiver performed well in terms of symbol error rate. The above research on VLC channels is very important to more deeply understand and improve the communication performance of these systems.

In comparison with the above studies, our work focuses on an LOS channel to study the visible light irradiation range in the LOS channel and to quantitatively analyze the volume share of the optical signal reception area in the communication channel. We simulated a VLC system with fixed positions at both ends of the transceiver, based on a theoretical LOS channel model [41,42,43]. We discuss the effects of five different indicators on the communication system, namely communication distance, transmitter half angle, irradiance half angle at the detector, irradiance half angle at the source, and detection surface area. We built a 10 m indoor LOS channel based on an IM/DD modulation VLC system with the position of the transmitter and the receiver not fixed. We established a 3D model of an entire communication channel using comparative experiments of different lampshade heights, and divided it into three different communication areas. Further analysis showed that when the transmitter angle was fixed, the longest communication distance varied non-linearly with the change in the lampshade height. At the same time, the volume ratio (VR) of areas I and III was close to 20∼30% and that of area II was the highest, reaching more than 50%, which was the best signal receiving area obtained.

The main contributions of this paper are as follows: (1) A three-dimensional BER distribution model of VLC under a conical lampshade was established; (2) The optimal signal reception position in the illuminated area was determined.

The rest of the article is organized as follows: Section 2 introduces the theoretical model of VLC; Section 3 illustrates the theoretical simulation and analysis of the VLC; Section 4 explains the VLC experiment setup, presents the experimental results, and provides a discussion. A summary of the work is given in Section 5.

## 2. Theoretical Model of VLC

While studying VLC channels, LOS and NLOS channels are generally also taken into consideration. When considering the LOS channel alone, the dc gain H(0) [44] of the communication system is calculated accurately [2]. When considering the diffuse reflection channel, the light at the transmitter needs to pass through the diffuse reflection of the wall or other objects to reach the receiver. Figure 1 shows the theoretical model of the VLC system considered herein.

In Figure 1, *d* is the distance between the transmitter and the receiver, and ϕ is the angle. The field of view (FOV) at the receiver is ψ. When the transmission power is $P_{t}$, the irradiance (W/cm^2^) is $I_{s}$(*d*,ϕ)=$P_{t}$$R_{o}$(ϕ)/$d^{2}$. Therefore, the dc gain of the channel is written as
(1)$$H{\left(0\right)}_{LOS}=\left\{\begin{array}{cc} \frac{A}{d^{2}}R_{o}\left(\phi \right)T_{s}\left(\psi \right)g\left(\psi \right)cos\psi , & 0\leq \psi \leq \Psi_{c}\\ 0 & \theta >\Psi_{c} \end{array}\right.$$
where H(0) is the dc gain and *A* is the area of the detector. $R_{o}$(ϕ) is the radiant intensity of the transmitter. $T_{s}\left(\psi \right)$ is the signal transmission of the filter [45]. g(ψ) is the concentration gain, and $\Psi_{c}$ is the maximum FOV of the receivable area (usually $\Psi_{c}$≤π/2). In Equation (Equation 1), $R_{o}$(ϕ) = [(m + 1)/2π]$cos^{m}$ϕ, where *m* is the transmitter semi-angle at half power and is written as $m=-ln2/ln(cos\Phi_{1/2})$. $\Phi_{1/2}$ is the transmitter semi-angle. By substituting these values, Equation (Equation 1) is rewritten as
(2)$$\begin{array}{c} H{\left(0\right)}_{LOS}=\left\{\begin{array}{cc} \frac{(m+1)A}{2\pi d^{2}}cos^{m}\phi T_{s}\left(\psi \right)g\left(\psi \right)cos\psi , & 0\leq \psi \leq \Psi_{c}\\ 0 & \theta >\Psi_{c} \end{array}\right. \end{array}$$
where the dc gain is directly proportional to the area of the detector *A* and inversely proportional to the distance $d^{2}$. Under certain conditions, the power efficiency of the LOS channel is maximized by increasing the radiation intensity of the transmitter $R_{o}$(ϕ). When the receiver moves, the changes in other parameters (d, $T_{s}\left(\psi \right)$ and g(ψ)) are compensated by changing the value of ϕ, thereby optimizing the communication efficiency of the communication system.

## 3. Theoretical Simulation and Analysis of VLC

### 3.1. Simulation Model

In this study, the theoretical model is designed using Optisystem simulation software. Numerical simulation is carried out based on Formulas (Equation 1) and (Equation 2). Figure 2 shows the entire simulation process.

At the transmitter of Figure 2, a series of binary sequences are generated by a pseudo-random bit sequence generator. This sequence is converted into a non-return-to-zero (NRZ) signal by an NRZ pulse generator. The signal is then modulated by a Mach–Zehnder modulator (the extinction ratio is 30 dB) and transmitted by a single LED (wavelength is 380–780 nm). At the receiver, an optical signal is converted into an electrical signal by a PIN photodetector (the responsivity is 1 A/W and the dark current is 10 nA). After being amplified by a transimpedance amplifier (open-loop voltage gain is 40 dB) and an automatic gain control amplifier (output voltage is 5 V), the electrical signal is filtered in a low-pass Bessel filter. The filtered signal is restored to its original NRZ signal by a 3R generator. Finally, the NRZ signal is analyzed using the BER analyzer. The experimental parameters are set as shown in Table 1.

We can infer from Table 1 that VLC with a BER of $4.6\times 10^{-5}$ is achieved at a distance of 1 m, which is lower than the forward error correction (FEC) limit of $3.8\times 10^{-3}$. The Q factor of this communication system is 3.58, and the eye height is 0.12, which indicates that the communication performance of the system is stable.

### 3.2. Simulation Results and Analysis

After analyzing the VLC model, we conducted a comparative analysis of various parameters that affect the communication performance based on the parameter settings in Table 1. Figure 3 shows the simulation results.

In Figure 3a, according to $\Psi_{c}$≤π/2 in Formula (Equation 1), the transmitter half angle is set to vary from 0° to 80°, with an interval of 2°, and the irradiance half angle at the source varies from 10° to 50°, with an interval of 10°. The irradiance half angle at the detector is 20°. The logarithmic value of the BER is reduced when the transmitter half angle is around 20° and increases from −33.68 to −22.12 when the irradiance half angle at the source increases from 10° to 50°. When the transmitter half angle is greater than or less than 20°, a large BER is generated. Therefore, in addition to considering the influence of the transmitter half angle and the irradiance half angle at the source, the irradiance half angle at the detector also needs to be considered in order to reduce the BER.

In Figure 3b, the irradiance half angle at the detector varies from 0.001° to 80°, with an interval of 2°, and that at the source varies from 10° to 50°, with an interval of 10°. When the irradiance half angle at the detector is 0.001°, the logarithmic value of the BER increases from −17.59 to −11.13, with the irradiance half angle at the source increasing from 10° to 50°. This indicates that the irradiance half angle at the source positively correlates with the BER.

In Figure 3c, the communication distance varies from 0.001 to 10 m, with an interval of 0.25 m. The transmitter half angle varies from 20° to 60°, with an interval of 10°. When the distance is 0.001 m, the logarithmic value of the BER increases from −43.13 to −25.85, with the transmitter half angle increasing from 20° to 60°. When the BER is at the FEC limit of $3.8\times 10^{-3}$, the communication distance increases from 2.44 to 3.97 m with the transmitter half angle decreasing from 60° to 20°. This indicates that the communication distance affects the communication quality of the communication system.

In Figure 3d, the detection surface area varies from 0.001 to 20 cm${}^{2}$, with an interval of 0.5 cm${}^{2}$. The communication distance varies from 2 to 10 m, with an interval of 2 m. When the communication distance is 2 m, the logarithmic value of the BER reaches the minimum value of −37.1, and the detection surface area is 11 cm${}^{2}$. When the detection surface area is larger than 11 cm${}^{2}$, the logarithmic value of the BER does not decrease further. This indicates that when the communication distance is fixed, the continuous increase in the detection surface area has a limited impact on reducing the BER.

Our analysis of the above indicators suggests that the minimum BER of the VLC system between the transmitter half angle and the irradiance half angle at the source is related to the irradiance half angle at the detector. Similarly, the obtainable communication distance increases as the transmitter half angle decreases and the concentration of the light increases.

However, the transmitter and the receiver need to be accurately aligned, which has many advantages for the communication system of the fixed transceiver, which is difficult to achieve in mobile transceiver systems. In particular, mobile transceiver systems require a large receiving space. Based on the above findings, in this study we increased the communication distance from 40 cm to 10 m using a lampshade. We built a 10 m VLC system indoors, and the receiver was arranged in such a way that it could be moved longitudinally and laterally on the track. The longitudinal distance was varied from 0 to 10 m. The lateral variation was varied from −20 cm to +20 cm. We ensured for a fixed transmitter that the receiver had enough receiving space.

## 4. VLC Experiment Setup and Discussion

### 4.1. Experiment Setup

In this study, we designed an experimental setup of VLC. In this communication system, an STM32F103C8T6 chip was used as the microcontroller unit (MCU), and a 10 m VLC system was achieved using the IM/DD technique. Figure 4 shows the entire transceiver system.

Figure 4 shows how the signal was transmitted from the computer through an RF wireless module A to B, where A and B had the same frequency. The wireless operating frequency band was 433.4–473.0 MHz. It had 100 channels, with a 400 kHz incremental step between each channel. The data were then transmitted to the MCU of a transmitting circuit board through a serial port. The communication rate of the transmitter was 115,200 bps. The received data were transmitted by a single LED after being modulated and amplified. The LED power was 3 W, and the luminous flux was 190–210 lm. The lampshade half angle was 30° and the LOS channel was 10 m. At the receiver, the optical signal was detected by a Hamamatsu S6801 photodetector with an effective active area of 150 mm${}^{2}$ and a half angle of 35°, and the data signal was amplified, demodulated, and transmitted to the MCU. The MCU in the receiver sent demodulated data to a wireless module C through the serial port. Finally, the data were sent from the wireless module C to D and displayed on the computer, where C and D had the same frequency. Figure 5 shows the experimental setup.

Figure 5a,b show photographs of the experimental scenarios when the lights are turned on and off, respectively. We can see from Figure 5a that the transmitter of the communication system was fixed and the receiver was moved longitudinally (10 m) and laterally (1 m) on the track. The transmitter and the receiver were at the same height (70 cm). We found from the analysis of the above communication indicators that use of the lampshade increased the communication distance significantly. Therefore, this study conducted a series of experiments to study the effect of the lampshade (the height was from 2 to 6 cm (Figure 5c) and the lampshade half angle was 30°) on the VLC performance. On the basis of the BER distribution of the communication system to determine the best position of the mobile receiver, the communication BER was recorded at 5 cm, 10 cm, 20 cm, 40 cm, 60 cm, 80 cm, 100 cm, 150 cm, 200 cm, 250 cm, 300 cm, 400 cm, 500 cm, 600 cm, 700 cm, 800 cm, 900 cm, and 1000 cm. The lateral distance varied from −20 cm to +20 cm, and one experiment was performed per 1 cm. Each experiment was repeated five times in order to ensure the reliability of the data. All experiments were conducted with the lights turned off to avoid the influence of fluorescent lamps and other light on the experiment.

### 4.2. Experimental Results

The experimental results are divided into three areas (I–III) based on the BER (see Figure 6, Figure 7, Figure 8, Figure 9 and Figure 10). In area I, the longitudinal distance varied from 5 to 150 cm, and changed in the BER were symmetrically distributed on both sides of the origin when the receiver moved laterally from −20 to +20 cm. When the longitudinal distance was 5 cm, the maximum lateral receiving distance in the negative direction of the receiver was −10 cm and that in the positive direction of the receiver was 9 cm (see Figure 7a), and therefore the lateral receiving distance of the receiver was 19 cm. When the longitudinal distance was 10 cm, the lateral receiving distance of area I reached its maximum when compared with other lampshade heights, and the distance was 25 cm. The lateral receiving distance became smaller with increasing longitudinal distance and reached its smallest value of 12 cm when the longitudinal distance was 150 cm. At the same time, there was an interval with a BER of 100% in the positive and negative directions of the origin. The primary reason for this is that the light reflected by the lampshade was superimposed, producing an inter-symbol interference at the receiver.

In area II, due to the differences in the heights of the lampshades, the variation range of the BER in the longitudinal direction also varied. When the lampshade height was 2 cm, the variation range of the longitudinal distance was at its minimum, from 200 to 400 cm (see Figure 6b). As the lampshade height increased, as shown in Figure 8b, the variation range of the longitudinal distance also reached its maximum, from 200 to 600 cm. After this, the variation range of the longitudinal distance remained unchanged. At the same time, the lateral receiving distance at the receiver increased with increasing longitudinal distance. When the longitudinal distance increased from 200 to 600 cm, the lateral receiving distance increased from 14 to 24 cm and from 17 to 31 cm, respectively (see Figure 8b and Figure 10b). In this area, the reflected light had little effect on communication performance, and the receiving range of the receiver increased with increasing lampshade height and longitudinal distance.

In area III, when the longitudinal distance increased, the light intensity attenuated rapidly and the lateral receiving range of the receiver decreased irregularly. When the longitudinal distance increased from 600 to 1000 cm, the receiving distance of the receiver decreased from 22 cm to 0 (see Figure 7c). At the same time, the longest receiving distance in the longitudinal direction was 923 cm. In this area, when the lampshade height increased from 2 to 6 cm, the longest communication distance of the receiver was increased from 725 to 1060 cm. The lampshade height was increased by 4 cm and the communication distance was increased by 46.34%. This provides evidence that the increase in the communication distance directly affects the system communication performance, and that an increase in lampshade height improves the system communication distance.

### 4.3. Establishment of 3D Model

To determine the best receiving area, we established an ideal illumination area model and an experimental data model using the above experimental data, the 3D schematic diagrams of which are shown in Figure 11.

In Figure 11a, ϕ is the FOV at the transmitter. The illumination model at this point is the conventional Lambert radiation model. In Figure 11b, the red solid line area indicates the theoretical signal reception area. When the receiver is located at any position in this area, the optical signal can theoretically be received. In Figure 11c, the red solid line area indicates the actual signal receivable area. Due to the influence of reflected light, the light sensitivity limitation of the sensor at the receiver, and other factors, the optical signal that can be received in the experiment is somewhat different from the theoretical situation. In Figure 11d, $d_{0}$ is the outer diameter of the lampshade; $d_{1}$, $d_{2}$, and $d_{3}$ are the longitudinal distances of the receiver in area I, area II and area III, respectively; and $L_{I}$, $L_{II}$, and $L_{III}$ are the lateral distances in area I, area II, and area III, respectively.

The received optical signal is error-free, as shown in Figure 11d. The interference factor in this area is light intensity attenuation caused by the change in the communication distance. $L_{I-1}$ represents the longitudinal distance when the lateral receiving distance of the optical signal is increased to the maximum for the first time. Area I is mainly affected by two aspects of interference: (i) the attenuation of the light intensity due to the change in the distance of the reflected light and (ii) the inter-symbol interference caused by the overlapping of the reflected light. Under the effect of two kinds of interference, the receivable area of the signal exhibits an irregular light distribution.

From Figure 11d, the longest distance index (LDI) of the VLC system is defined as
(3)$$\begin{array}{cc} LDI=\underset{j}{max}\sum L_{i}, & (i=I,II,III;j=2,3,4,5,6). \end{array}$$

The longest lateral distance (LLD) is expressed as
(4)$$\begin{array}{cc} LLD=\underset{i,j}{max}d_{i}, & (i=1,2,3;j=2,3,4,5,6). \end{array}$$

Table 2 shows the experimental data of different lampshade heights.

We can infer from Table 2 that as the lampshade height increased, the value of $d_{1}$ changed slightly, indicating that the maximum horizontal receiving distance of the optical signal in area I was not highly correlated with the lampshade height.

With the increase in the lampshade height, the value of $L_{I-1}$ first increased and then stabilized, indicating that the reflected light had a significant impact on optical communication when the receiver and the transmitter were very close. With increasing longitudinal distance, the effects of reflected light gradually diminished and eventually became stable. The values of $d_{2}$, $L_{I}$, and $d_{3}$ increased with increasing lampshade height, indicating that the change in the lampshade height promoted the longitudinal and the horizontal receiving distance of the optical signal in area I. When the lampshade height increased, $L_{II}$ first increased and then stabilized, indicating that the change in the lampshade height could not further increase the longitudinal distance of area II. At this time, the light intensity attenuation caused by the change in the distance between the transmitter and the receiver had a significant effect on communication. When the lampshade height increased, the value of $L_{III}$ did not change significantly. The main reason for this is that the light intensity of area III was seriously attenuated, and the effect of external environment was prominent at this time. Therefore, using LDI and LLD as the evaluation standard of the system, the lampshade height was 6 cm, LDI = 1060 cm, and LLD = 31 cm. This is the optimal parameter setting configuration. Analysis of Figure 11d and Table 2 indicates that when the volume ratio (VR) of the signal receiving area is used as the evaluation index of the VLC system, the maximum VR is expressed as
(5)$$\begin{array}{cc} VR=\underset{i,j}{max}\frac{V_{i}}{\sum V_{i}}\times 100\%,\; & (i=I,II,III;j=2,3,4,5,6), \end{array}$$
where $V_{i}$ is the volume of the signal receivable regions in areas I–III. From the above analysis, the volumes of the signal receivable range of areas I–III were calculated and are shown in Figure 12.

It can be seen from Figure 12a that the increase in the lampshade height had a certain enhancement effect on the entire communication range. Considered alongside the data analysis in Table 2, it can be deduced that area II was least affected by the external environment in the entire area that can receive signals. We extracted the VR of area II occupying the entire communication area, and Figure 12b was obtained.

The volume proportion of area II gradually increased with increasing lampshade height and then decreased (see Figure 12b). This shows that the continuous increase in the lampshade height could not increase the proportion of area II continuously. For different lampshade heights, the volume proportion of area II in the entire signal receivable area was greater than 50%, which indicates that area II was the main communication area in the entire communication range. When the lampshade height was 5 cm, VR was 58.8%, which indicates that the signal stable area in the signal receiving range of VLC was the largest. To further increase the volume of area II, methods such as increasing the transmitting power and improving the light detection sensitivity at the receiver end could also be used to obtain a more stable and larger communication area.

### 4.4. Results Discussion

This paper describes our investigation of the effect of lampshade height on VLC. The light irradiation area was divided into three different signal receiving areas according to the BER distribution. We experimentally obtained the volume ratio of the different receiving areas in the entire illumination area, and modeled the optimal area for VLC mathematically accordingly. Following the results of this study, the effect of light reflection from the ground on BER needs to be considered in subsequent experiments. Meanwhile, the gradual weakening of the light in the edge part of the light illumination caused the BER to change to 100% in a very short distance. How to improve the detection ability of weaker light signals needs to be studied in future work. For visible light communication, three parts are generally studied: the transmitter, the communication channel, and the receiver. Since the focus of this paper is on the communication channel aspect, a more efficient modulation and demodulation method is used at the transmitter side: the use of LED devices with higher luminous flux, and the enhancement of the sensing performance of the photoelectric sensor for weak light at the receiving end.All of these strategies can improve the communication distance of the channel and reduce the BER of the communication to a certain extent.

## 5. Summary

In this paper, we discuss the effect of communication distance, transmitter half angle, irradiance half angle at the detector, irradiance half angle at the source, and detection surface area on the communication system. To study the effect of the change in the lampshade height on the VLC system, a 10 m LOS channel indoors was built using the IM/DD technique and a single white LED. In the experiment we creatively divided the entire communication range into three different optical signal receiving areas, and analyzed each area. The experimental results showed that area I was mainly affected by the reflected light at a short distance. Area II was a stable communication area whose communication range was positively related to the lampshade height. Area III was mainly affected by the communication distance and external environmental interference. The volume of area II in the entire optical signal receivable area accounted for more than 50%.

Our future work will focus on communication stability issues in areas I and III. We may consider adding signal amplifiers at the transmitter side for improving signal strength. It would also be interesting to add filters in front of the receiver to reduce the effect of background light on visible light communications, and to use more sensitive photoelectric sensors to enhance the detection of faint light at the receiver side.

The above methods could improve the communication performance of the whole system. This study has certain guiding significance for studying the distribution of optical signals and the improvement of communication performance in VLC.

## Figures and Tables

**Figure 1 sensors-23-00094-f001:**
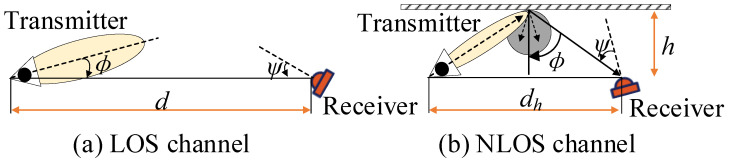
Theoretical model. Here *d* is the distance between the transmitter and the receiver and ϕ is the angle; ψ is the field of view (FOV) at the receiver; *h* is the distance between the receiver and the reflecting surface; and $d_{h}$ is the distance between the transmitter and the receiver.

**Figure 2 sensors-23-00094-f002:**
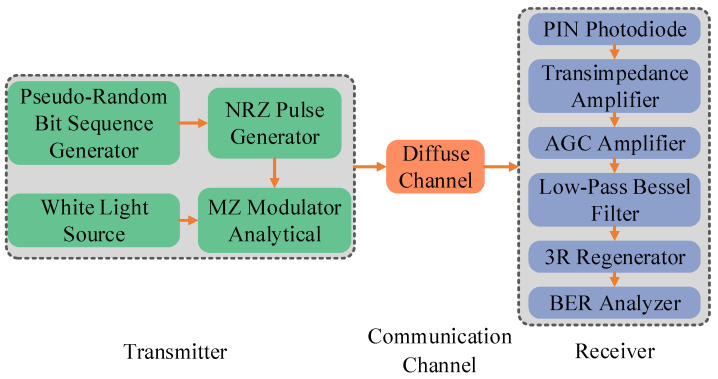
Simulation diagram.

**Figure 3 sensors-23-00094-f003:**
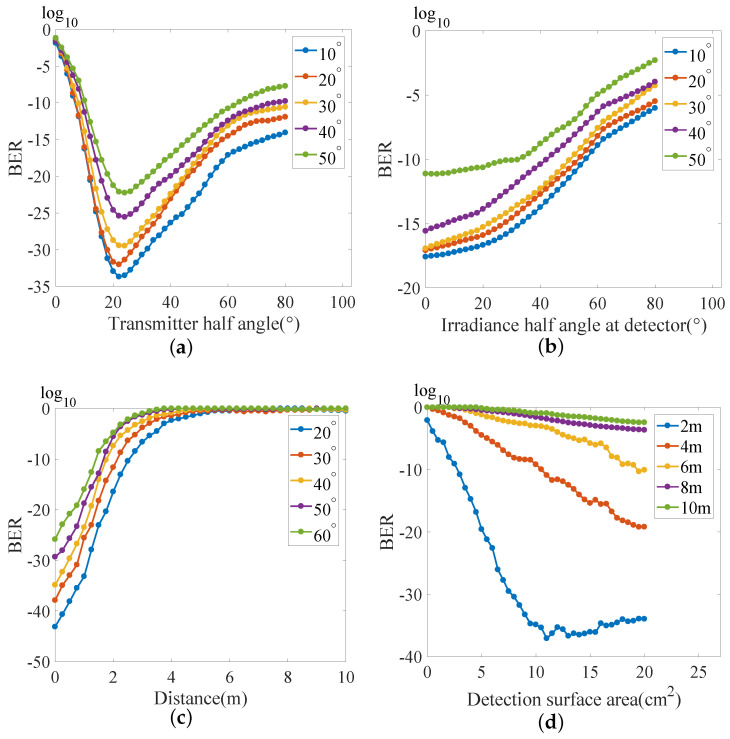
Theoretical simulation results. The relationships between: (**a**) the irradiance half angle at the source and the transmitter half angle; (**b**) the irradiance half angle at the source and the irradiance half angle at the detector; (**c**) the transmitter half angle and communication distance; and (**d**) communication distance and detection surface area.

**Figure 4 sensors-23-00094-f004:**
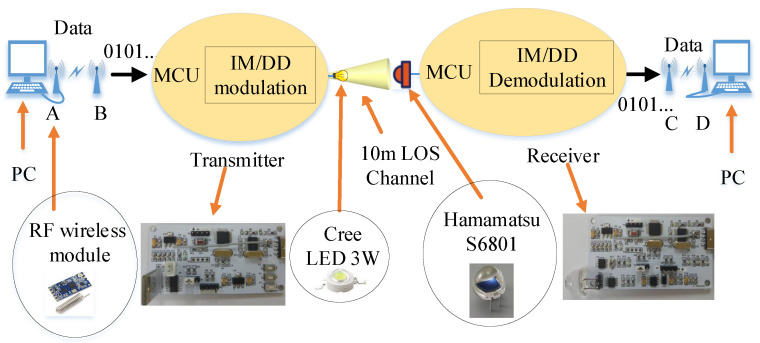
Transceiver system.

**Figure 5 sensors-23-00094-f005:**
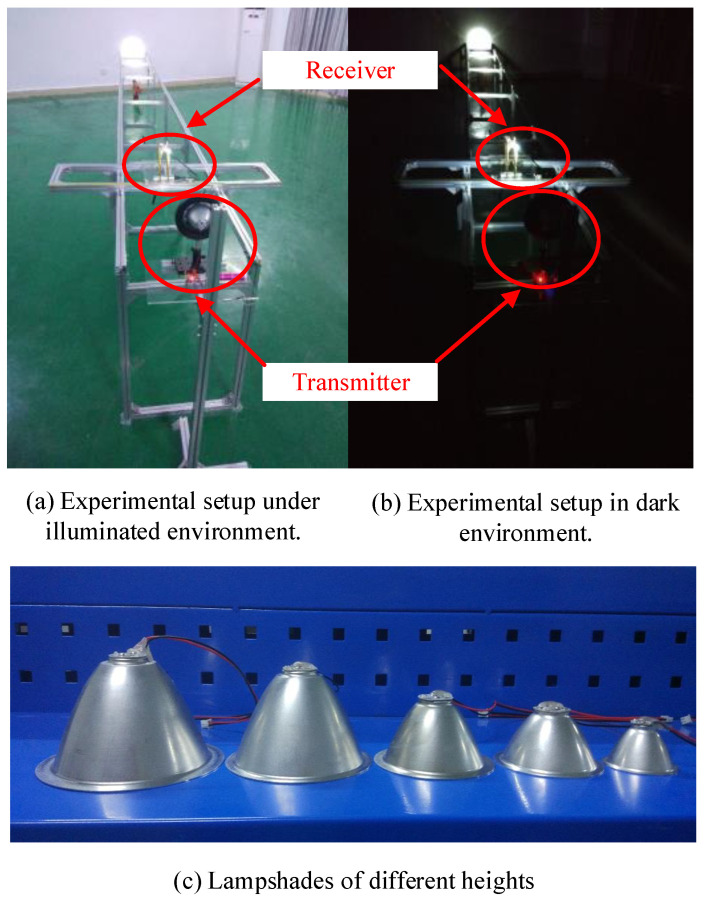
Experimental setup.

**Figure 6 sensors-23-00094-f006:**
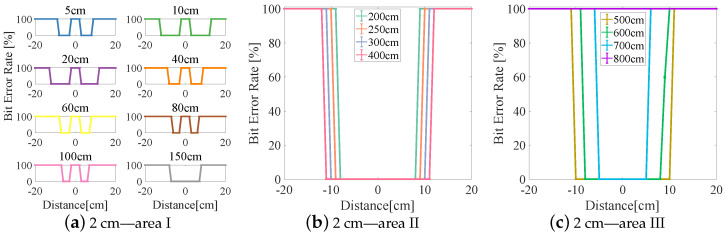
Lampshade height: 2 cm.

**Figure 7 sensors-23-00094-f007:**
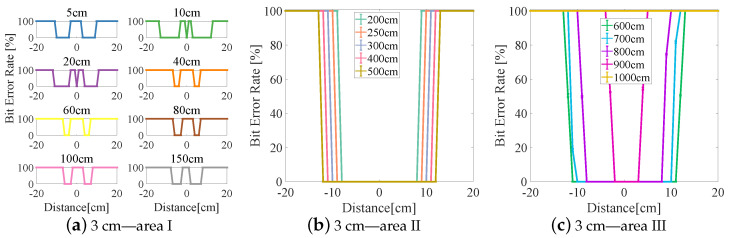
Lampshade height: 3 cm.

**Figure 8 sensors-23-00094-f008:**
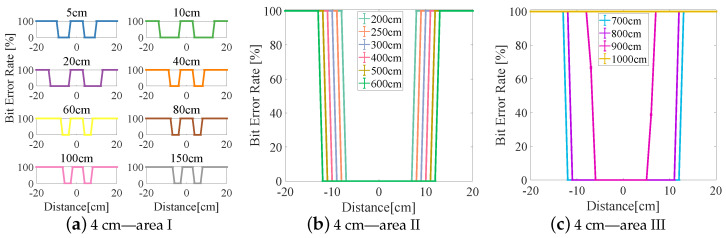
Lampshade height: 4 cm.

**Figure 9 sensors-23-00094-f009:**
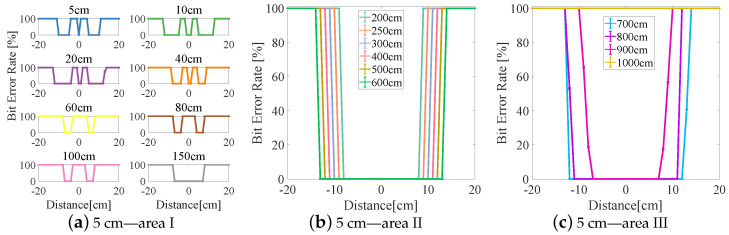
Lampshade height: 5 cm.

**Figure 10 sensors-23-00094-f010:**
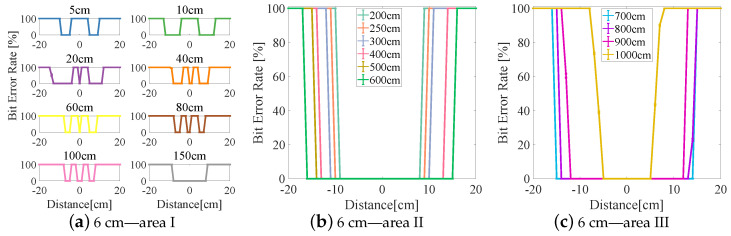
Lampshade height: 6 cm.

**Figure 11 sensors-23-00094-f011:**
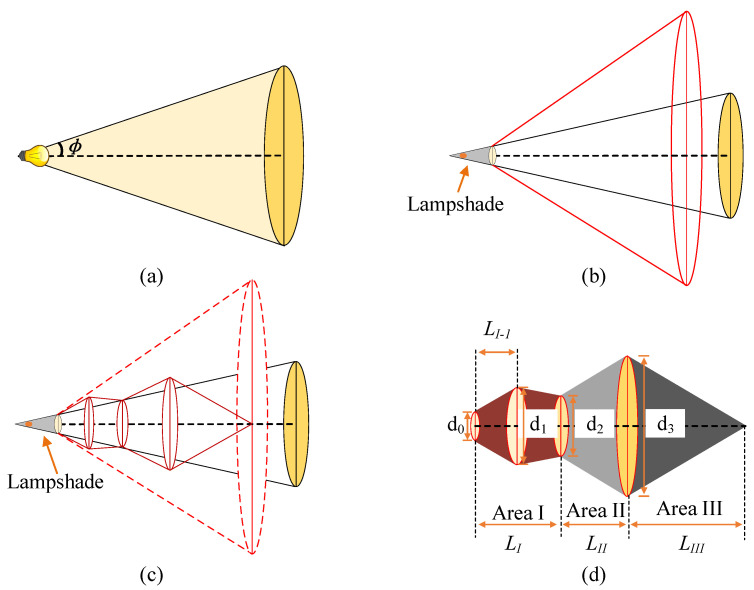
Experimental model. (**a**) Model of light radiation without lampshade. (**b**) Light reflection model with lampshade. (**c**) Receivable region of the optical signal for the reflection model. (**d**) Optical signal error-free distribution area.

**Figure 12 sensors-23-00094-f012:**
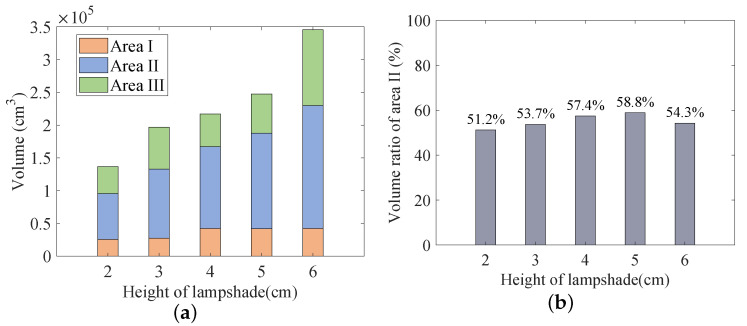
(**a**) Volume distribution of signal receivable area. (**b**) Volume ratio of area II.

**Table 1 sensors-23-00094-t001:** The simulation parameters.

Parameters	Values
Transmitter half angle (°)	60
Irradiance half angle at detector (°)	20
Irradiance half angle at source (°)	20
Communication distance (m)	1
Detection surface area (cm${}^{2}$)	1

**Table 2 sensors-23-00094-t002:** Experimental data.

Lampshade Height (cm)	2	3	4	5	6
d${}_{1}$ (cm)	24	25	25	24	24
$L_{I-1}$ (cm)	10	10	20	20	20
d${}_{2}$ (cm)	12	12	13	14	14
$L_{I}$ (cm)	100	100	150	150	150
d${}_{3}$ (cm)	22	24	24	26	31
$L_{I}$+$L_{II}$ (cm)	400	500	600	600	600
$L_{I}$ + $L_{II}$ + $L_{III}$ (cm)	725	923	930	938	1060

## Data Availability

Not applicable.

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
