# Peer review of "Three-Dimensional Division of Visible Light Communication Irradiation Area"

_sensors, 2022, doi:10.3390/s23010094_

Round 1
Reviewer 1 Report
The paper analyses the quality of VLC (white light diode) in a 10m LOS channel with various settings. While the topic is certainly
The experimental setup and results are presented well. A minor note would be that the experiment conditions include ambient lights turned off, but beyond this there could be a further discussion or set of experiments treating the problem of various sources (different intensities and/or spectral composition) for the ambient light, as one possible generalization of the experiment. Or a paragraph arguing what possible scenarios would never have the problem of interfering ambient light. On this note, the conclusions could be amended with the description of the limitations and constraints of the described setups.
The use of English is excellent, with only the occasional awkward phrasing such as:
line 40: "In this work, the main contribution of the article" is somewhat redundant.
line 120-121: repeated use of the word "communication" (thrice) in the same sentence, probably in an effort to avoid any misunderstanding.
Author Response
Response to Reviewer 1 Comments
Point 1: A minor note would be that the experiment conditions include ambient lights turned off, but beyond this there could be a further discussion or set of experiments treating the problem of various sources (different intensities and/or spectral composition) for the ambient light, as one possible generalization of the experiment. Or a paragraph arguing what possible scenarios would never have the problem of interfering ambient light. On this note, the conclusions could be amended with the description of the limitations and constraints of the described setups.
Response 1: Thanks for your comment. We have added text descriptions in the summary section of the article. “Our future work will focus on communication stability issues in areas I and III. We may consider adding signal amplifiers at the transmitter side for improving signal strength. Adding filters in front of the receiver to reduce the effect of background light on visible light communications. Using more sensitive photoelectric sensors to enhance the detection of faint light at the receiver side. The above methods can improve the communication performance of the whole system to some extent.” We hope we answered the reviewer’s comment correctly and clearly. And please let us know if we need to provide a clearer response to this comment.
Point 2: Line 40: "In this work, the main contribution of the article" is somewhat redundant.
Response 2: We agree with the reviewer. We have truncated the expression here and resummarized the article's contribution. “The main contributions of this paper are as follows: (1) A three-dimensional BER distribution model of VLC under a conical lampshade was established; (2) The optimal signal reception position in the illuminated area was determined.” We hope we answered the reviewer’s comment correctly and clearly. And please let us know if we need to provide a clearer response to this comment.
Point 3: Line 120-121: repeated use of the word "communication" (thrice) in the same sentence, probably in an effort to avoid any misunderstanding.
Response 3: Thanks for your comment. We agree with the review’s comment. Our modification is as follows: “It shows that the transmission distance of the signal affects the communication quality of the system.” We hope we answered the reviewer’s comment correctly and clearly. And please let us know if we need to provide a clearer response to this comment.

Reviewer 2 Report
The introduction section must be extended.
The paper structure is absent at the end of the introduction.
The motivation and main contribution must be highlighted.
Revise all explanations of all mathematical variables.
Comparative study with related works must be included.
More recent related works must be included.
Improve the discussion section.
Add more future suggestions.
Author Response
Response to Reviewer 2 Comments
Point 1: The introduction section must be extended.
Response 1: Thanks for your comment. We have included the following text description in the introduction section of the article.
“The communication channel is an important part of the whole VLC system. Related studies have shown that the improvement of communication channels helps to enhance the communication performance of the system. The LOS channel and non-LOS (NLOS) channel are analyzed in Literature [33-35]. Mohanna et al. [33] analyzed the effect of reflected light from walls on the receiver side by simulating the LOS channel and NLOS channel. By arranging multiple LEDs in both LOS and NLOS channels, Priyanka et al. [34] improves the uniformity of light illumination within the communication channel and mitigates the effect of inter-symbol interference. Long [35] researched the performance evaluation of VLC system with various FOV, elevation and azimuth receivers by simulating the LOS and NLOS models.A mixed-integer linear programming model was developed.
At the same time, some researchers have turned their attention to VLC channels in different application scenarios. Sc et al. [36] introduced a VLC channel for I2V communication. The corresponding propagation model was established by linear regression. The proposed model shows higher accuracy than the traditional Lambertian model. Ding et al. [37] investigated the VLC channel characteristic evolution from the conventional Lambertian beam to typical non-Lambertian beams. Studies have shown that Side Emitter beams provide a more uniform capacity distribution in a distributed transmitter configuration. Zhu et al. [38] proposed a three dimensional space-time-frequency non-stationary geometry based stochastic model for indoor VLC channels. The accuracy and practicality of the model are verified by simulation experiments. Palacios et al. [39] proposed a hemispherical three-dimensional dust particle distribution model based on an underground mining scenario. By comparing with other models, the proposed model is more accurate and realistic in terms of metrics assessment. Li et al. [40] constructed a neural network-based transceiver for compensating the effect of variable inter-symbol-interference in VLC system. The results show that the neural network-based transceiver performs well in terms of symbol error rate. The above research on VLC channel is very important to understand more deeply and improve the communication performance of the system.”
“In comparison with the above studies, our work focuses on the LOS channel to study the visible light irradiation range in the LOS channel and to quantitatively analyze the volume share of the optical signal reception area in the communication channel.”
“The main contributions of this paper are as follows: (1) A three-dimensional BER distribution model of VLC under a conical lampshade was established; (2) The optimal signal reception position in the illuminated area was determined.”
“The rest of the article is organized as follows: section 2 introduces the theoretical model of VLC; section 3 illustrates the theoretical simulation and analysis of VLC; section 4 explains VLC experiment setup and discussion; relevant experiments and discussions are presented in section 5; section 6 provides a summary of this work.” We hope we answered the reviewer’s comment correctly and clearly. And please let us know if we need to provide a clearer response to this comment.
Point 2: The paper structure is absent at the end of the introduction.
Response 2: Thanks for your comment. We have added text descriptions to the paper structure at the end of the introduction. “The rest of the article is organized as follows: section 2 introduces the theoretical model of VLC; section 3 illustrates the theoretical simulation and analysis of VLC; section 4 explains VLC experiment setup and discussion; relevant experiments and discussions are presented in section 5; section 6 provides a summary of this work.” We hope we answered the reviewer’s comment correctly and clearly.
Point 3: The motivation and main contribution must be highlighted.
Response 3: We agree with the reviewer. We have resummarized the article's contribution. “The main contributions of this paper are as follows: (1) A three-dimensional BER distribution model of VLC under a conical lampshade was established; (2) The optimal signal reception position in the illuminated area was determined.” We hope we answered the reviewer’s comment correctly and clearly. And please let us know if we need to provide a clearer response to this comment.
Point 4: Revise all explanations of all mathematical variables.
Response 4: Thanks for your comment. We agree with the reviewer. We have removed the previous Fig. 11 and modified the original Fig. 12. A description of the modified figure has been added to the article. “As shown in Fig. 11(a), Ï• is the FOV at the transmitter. The illumination model at this point is the conventional Lambert radiation model. In Fig. 11(b), the red solid line area indicates the theoretical signal reception area. When the receiver is located at any position in this area, the optical signal can be received theoretically. In Fig. 11(c), the red solid line area indicates the actual signal receivable area. Due to the influence of reflected light, the light sensitivity limitation of the sensor at the receiver and other factors, the optical signal that can be received in the experiment is somewhat different from the theoretical situation. In Fig. 11(d), d0 is the outer diameter of the lampshade. d1, d2 and d3 are the longitudinal distances of the receiver in Area I, Area II and Area III, respectively. LI , LI I and LI I I are the lateral distances in Area I, Area II and Area III, respectively.” We hope we answered the reviewer’s comment correctly and clearly. And please let us know if we need to provide a clearer response to this comment.
Point 5: Comparative study with related works must be included.
Response 5: Thanks for your comment. We have added a text description in the introduction part of the article. “In comparison with the above studies, our work focuses on the LOS channel to study the visible light irradiation range in the LOS channel and to quantitatively analyze the volume share of the optical signal reception area in the communication channel.” We hope we answered the reviewer’s comment correctly and clearly. And please let us know if we need to provide a clearer response to this comment.
Point 6: More recent related works must be included.
Response 6: Thanks for your comment. We have added a text description in the introduction part of the article. “The communication channel is an important part of the whole VLC system. Related studies have shown that the improvement of communication channels helps to enhance the communication performance of the system. The LOS channel and non-LOS (NLOS) channel are analyzed in Literature [33-35]. Mohanna et al. [33] analyzed the effect of reflected light from walls on the receiver side by simulating the LOS channel and NLOS channel. By arranging multiple LEDs in both LOS and NLOS channels, Priyanka et al. [34] improves the uniformity of light illumination within the communication channel and mitigates the effect of inter-symbol interference. Long [35] researched the performance evaluation of VLC system with various FOV, elevation and azimuth receivers by simulating the LOS and NLOS models.A mixed-integer linear programming model was developed.
At the same time, some researchers have turned their attention to VLC channels in different application scenarios. Sc et al. [36] introduced a VLC channel for I2V communication. The corresponding propagation model was established by linear regression. The proposed model shows higher accuracy than the traditional Lambertian model. Ding et al. [37] investigated the VLC channel characteristic evolution from the conventional Lambertian beam to typical non-Lambertian beams. Studies have shown that Side Emitter beams provide a more uniform capacity distribution in a distributed transmitter configuration. Zhu et al. [38] proposed a three dimensional space-time-frequency non-stationary geometry based stochastic model for indoor VLC channels. The accuracy and practicality of the model are verified by simulation experiments. Palacios et al. [39] proposed a hemispherical three-dimensional dust particle distribution model based on an underground mining scenario. By comparing with other models, the proposed model is more accurate and realistic in terms of metrics assessment. Li et al. [40] constructed a neural network-based transceiver for compensating the effect of variable inter-symbol-interference in VLC system. The results show that the neural network-based transceiver performs well in terms of symbol error rate. The above research on VLC channel is very important to understand more deeply and improve the communication performance of the system.” We hope we answered the reviewer’s comment correctly and clearly. And please let us know if we need to provide a clearer response to this comment.
Point 7: Improve the discussion section.
Response 7: Thanks for your comment. We have added a text description in the discussion part of the article. “For visible light communication, three parts are generally studied: the transmitter, the communication channel, and the receiver. Since the focus of this paper is on the communication channel aspect. Therefore, a more efficient modulation and demodulation method is used at the transmitter side; the use of LED devices with higher luminous flux; and the enhancement of the sensing performance of the photoelectric sensor for weak light at the receiving end. All of them can improve the communication distance of the channel and reduce the BER of the communication to a certain extent.” We hope we answered the reviewer’s comment correctly and clearly. And please let us know if we need to provide a clearer response to this comment.
Point 8: Add more future suggestions.
Response 8: Thanks for your comment. We have added text descriptions in the summary section of the article. ” Our future work will focus on communication stability issues in areas I and III. We may consider adding signal amplifiers at the transmitter side for improving signal strength. Adding filters in front of the receiver to reduce the effect of background light on visible light communications. Using more sensitive photoelectric sensors to enhance the detection of faint light at the receiver side. The above methods can improve the communication performance of the whole system to some extent.” We hope we answered the reviewer’s comment correctly and clearly. And please let us know if we need to provide a clearer response to this comment.

Round 2
Reviewer 2 Report
The revised paper is suggested to be accepted after some proofreading.